# Drug Repurposing for COVID-19 Treatment by Integrating Network Pharmacology and Transcriptomics

**DOI:** 10.3390/pharmaceutics13040545

**Published:** 2021-04-14

**Authors:** Dan-Yang Liu, Jia-Chen Liu, Shuang Liang, Xiang-He Meng, Jonathan Greenbaum, Hong-Mei Xiao, Li-Jun Tan, Hong-Wen Deng

**Affiliations:** 1Laboratory of Molecular and Statistical Genetics, College of Life Sciences, Hunan Normal University, Changsha 410081, China; danyang.liu@foxmail.com; 2Center for System Biology, Data Sciences, and Reproductive Health, School of Basic Medical Science, Central South University, Changsha 410013, China; ljch1999@csu.edu.cn (J.-C.L.); 196501012@csu.edu.cn (S.L.); xhmeng2020@csu.edu.cn (X.-H.M.); hmxiao@csu.edu.cn (H.-M.X.); 3Tulane Center of Biomedical Informatics and Genomics, Deming Department of Medicine, Tulane University School of Medicine, New Orleans, LA 70112, USA; jgreenb8@tulane.edu

**Keywords:** SARS-CoV-2, COVID-19, drug repurposing, network-based pharmacology

## Abstract

Since coronavirus disease 2019 (COVID-19) is a serious new worldwide public health crisis with significant morbidity and mortality, effective therapeutic treatments are urgently needed. Drug repurposing is an efficient and cost-effective strategy with minimum risk for identifying novel potential treatment options by repositioning therapies that were previously approved for other clinical outcomes. Here, we used an integrated network-based pharmacologic and transcriptomic approach to screen drug candidates novel for COVID-19 treatment. Network-based proximity scores were calculated to identify the drug–disease pharmacological effect between drug–target relationship modules and COVID-19 related genes. Gene set enrichment analysis (GSEA) was then performed to determine whether drug candidates influence the expression of COVID-19 related genes and examine the sensitivity of the repurposing drug treatment to peripheral immune cell types. Moreover, we used the complementary exposure model to recommend potential synergistic drug combinations. We identified 18 individual drug candidates including nicardipine, orantinib, tipifarnib and promethazine which have not previously been proposed as possible treatments for COVID-19. Additionally, 30 synergistic drug pairs were ultimately recommended including fostamatinib plus tretinoin and orantinib plus valproic acid. Differential expression genes of most repurposing drugs were enriched significantly in B cells. The findings may potentially accelerate the discovery and establishment of an effective therapeutic treatment plan for COVID-19 patients.

## 1. Introduction

The severe acute respiratory syndrome coronavirus 2 (SARS-CoV-2) caused the coronavirus disease 2019 (COVID-19) and triggered the largest pandemic since 1918 [1], which was responsible for >100 million cases and >2 million deaths reported globally [2]. However, there are no specific antiviral drugs for SARS-CoV-2 infection so far, for the reduction of morbidity and mortality of COVID-19, active symptomatic support was urgently needed [3].

According to recent reports [4,5,6], the majority of COVID-19 patients are currently given antiviral and antibiotic treatments or combination therapy including oseltamivir, ribavirin, lopinavir, ritonavir, and moxifloxacin. Additionally, several drugs are under clinical trials to verify their safety and efficacy for COVID-19 treatment, such as favipiravir, remdesivir, and hydroxychloroquine [7]. However, existing therapeutic options for the treatment of COVID-19 remain controversial. For example, remdesivir is an FDA Emergency Use Authorization (not FDA-approval) viral RNA polymerase inhibitor which has been widely used in COVID-19 patients [8], however, a recent randomized clinical trial demonstrated there was no significant beneficial effect [9]. Similarly, the COVID-19 WHO SOLIDARITY trial showed that other proposed treatments such as hydroxychloroquine, lopinavir, and interferon regimens appeared to have little or no effect on hospitalized COVID-19 patients [10]. Therefore, there is an urgent necessity to develop novel potential candidates for COVID-19 treatment.

Traditional drug development is a time-consuming and costly process that frequently takes 10–15 years and costs about 2–3 billion dollars from initial lab-scale experiments through the three phases of clinical trials and final approval for clinical usage [11]. Drug repurposing, as an effective and rapid drug discovery strategy from existing drugs [11,12], is considered the most practical approach as a rapid response to the emergent pandemic since the candidate treatments have already previously been tested for their safety [13]. The availability of the genomic sequence of SARS-CoV-2 has rapidly accelerated the development of clinical perspectives and recommendations. For example, David E. Gordon et al. identified 332 SARS-CoV-2 human protein-protein interactions and 69 drug candidates including 29 FDA-approved drugs, 12 clinical trial drugs, and 28 drugs at a preclinical stage [14]. Additionally, gene set enrichment analysis (GSEA) can be applied to identify underlying pathological processes using gene expression of COVID-19 patients, which can retrieve efficient drugs from patient-derived gene expression data using drug–target gene sets [15]. Therefore, the application of GSEA for drug targets based on drug–transcriptome-responses datasets and disease-associated gene sets can serve as an excellent screening tool for diseases that lack a safe and reliable cellular model for in vitro screening, such as COVID-19 [16].

This study uses an integrated network-based pharmacologic and transcriptomic approach to screen drug candidates for COVID-19 treatment. Network-based pharmacology is an effective and holistic tool to identify drug treatments, where the drug effects are provided by the distance between drugs and disease in the interactome [17]. Additionally, several databases containing genome-wide expression profiles of human cell lines treated with bioactive compounds have been developed for drug discovery [18]. Transcriptional profiling studies have successfully identified potential therapies for diseases such as breast cancer [19], diabetes [20], and Parkinson’s [21]. Using a network-based pharmacology approach combined with the transcriptional profiling databases, we detected 18 single drug candidates (e.g., dexamethasone, chloroquine, and tretinoin) and 30 synergistic drug combinations as potential therapies for COVID-19.

## 2. Materials and Methods

We screened novel drug combinations for COVID-19 by integrated network-based pharmacology and transcriptome analysis based on the following steps: (1) collection of COVID-19 related genes; (2) collection of target-available drugs and construction of drug–target modules; (3) calculation of network-based proximity between drug–target modules and COVID-19 related genes; (4) filtering drugs based on gene set enrichment analysis (GSEA); (5) network-based prediction of drug combinations (Figure 1). These steps will be detailed in the following.

### 2.1. Genes Related to COVID-19

Genes related to COVID-19 were retrieved from the latest SARS-CoV-2 human host data and a single-cell transcriptomic study of the peripheral immune response to severe COVID-19 (GSE150728). SARS-CoV-2 protein sequences, viral genomes, literature, clinical resources submitted to the National Center for Biotechnology Information (NCBI) on the SARS-CoV-2 special subject have been rapidly evolving [22]. In total, 65 SARS-CoV-2 human host proteins were selected from the coronavirus genomes of NCBI datasets and 1070 potential COVID-19 related genes were obtained from the transcriptomic study by selecting the differential expression genes (DEGs) between individual COVID-19 samples (*n* = 7) and healthy controls (*n* = 6) in 7 cell types, that was 409 genes from CD14+ Monocytes, 257 genes from CD16+ Monocytes, 261 genes from Dendritic Cells, 173 genes from NK (nature killer) cells, 180 genes from CD8+ T cells, 172 genes from CD4+ T cells and 481 genes from B cells (Appendix A) [23]. All the identified proteins were mapped to the official gene symbols of humans reported by the HUGO Gene Nomenclature Committee (HGNC). Finally, 63 SARS-CoV-2 related genes derived from human host proteins and 971 DEGs were retained as the COVID-19 potential related genes after removing duplicates.

Gene Ontology (GO) enrichment analysis was performed on the potential COVID-19 related genes to identify significant pathways. By using the R package ClusterProfiler [24], all potential COVID-19 related genes were functionally categorized according to their biological processes, cellular components, and molecular functions. Functional term enrichment analysis was performed to provide insights into the biological mechanisms underlying the COVID-19 related genes. Using this approach, only genes involved in the significantly enriched GO terms (*p*-value < 0.05) were retained for further analysis as COVID-19 related genes in the context of networks.

### 2.2. Drug–Target Relationship Modules

The drug information was obtained from DrugBank and SuperTarget [25,26]. Briefly, 7485 drugs with 21,335 drug–protein links were selected from DrugBank (version 5.1.6), and 3138 drugs with 16,579 drug–protein links were retrieved from SuperTarget. After removing drugs without targets as well as duplications, and converting all target genes into human gene symbols, 31,139 interactions containing 3121 targets of 7811 drugs were finally identified (Appendix A). A drug–target relationship module was defined by the drug–target interaction information, where multiple targets share one drug.

### 2.3. Network-Based Proximity between Drugs and COVID-19

A network-based approach was used to analyze the correlation between drug and disease, in which proximity scores were quantified by calculating the closest distance between the drug–target module and COVID-19 related genes in the context of the human protein-protein interaction (PPI) network. The PPI data were obtained from Pathway Commons (version 12), which contains over 5772 pathways and 2.4 million interactions [27]. Genes (nodes) with interaction (links) constructed a network graph of PPI, while the interaction between two nodes was undirected and unweighted. Here, a proximity score was defined by the average shortest path length between the drug target genes and their nearest disease proteins in the context of PPI to quantify the therapeutic effect of drugs [28,29]. Given the set of COVID-19 related genes sourced from SARS-CoV-2 proteins (S), the group of drug target genes (T), the shortest distance between two genes in the PPI network $d\left(s,t\right)$ where s∈S and t∈T (Equation (1)),
(1)$$d\left(S,T\right)=\frac{1}{\left|T\right|}\underset{t\in T}{\sum }min_{s\in S}\left(d\left(s,t\right)+w\right)$$ where *w* is the drug influencing weight, defined as $w=-\ln\left(D+1\right)$ if the drug target is one of the COVID-19 related genes sourced from DEGs (D is the connectivity degree of targets) and *w* = 0 otherwise.

A simulated reference distance score distribution corresponding to the drug was generated to assess the significance of the results by linking the drug’s random target modules and disease-related genes. Referenced drug modules were constructed by selecting random genes (denoted as R) with the same degree of drug target sets in the network, where the distance $d\left(S,R\right)$ indicates the relationship between a simulated drug and COVID-19. The reference distribution was established based on 30,000 replications. A drug with a score lower than 98% of the reference distribution scores was considered significant [28]. The network proximity was converted to Z-score based on permutation tests (Equation (2)):(2)$$Z\left(S,T\right)=\frac{d\left(S,T\right)-\mu_{d\left(S,R\right)}}{\sigma_{d\left(S,R\right)}}$$
where $\mu_{d\left(S,R\right)}$ and $\sigma_{d\left(S,R\right)}\;$ are the mean and standard deviation of the permutation tests.

### 2.4. Biological Enrichment Analysis of COVID-19 Related Genes on the Drug-Induced Expression Profiles

We performed GSEA as a further prioritization strategy to screen drug candidates by examining the distribution of disease-related genes in drug-induced gene expression profiles. GSEA was utilized to determine whether a priori defined sets of genes showed statistically significant enrichment in a collected gene list [30], which could identify whether drug candidates affected the expression of disease pathways. We first collected perturbation-driven gene expression profiles from LINCS (Library of Integrated Network-based Cellular Signatures), which provided transcriptional responses of human cells to chemical and genetic perturbation [31]. Human myeloid leukemia mononuclear (THP-1) cell line from blood was selected due to the important association of peripheral blood and myelomonocytic cells with COVID-19 [32,33,34]. The goal of GSEA was to determine whether the COVID-19 related genes sourced from the SARS-CoV-2 related gene set was randomly distributed throughout the drug-induced expression data set sorted by correlation with the phenotype of interest or enriched at either the top or bottom. Drugs with FDR (False Discovery Rate) less than 0.25 and ES (Enrichment Scores) higher than 0 were identified as potential drug candidates for COVID-19.

### 2.5. GSEA Analysis of Repurposing Drugs in Specific Cell-Types

According to the Seurat data provided by Aaron J. Wilk [23], we chose “Cell type (coarse)” as the standard to select scRNA seq data of seven cell types, including B Cells, CD14+ Monocytes, CD16+ Monocytes, CD4+ T Cells, CD8+ T Cells, Dendritic Cells, NK (natural killer) Cells, and calculated differentially expressed genes between total COVID-19 samples (*n* = 7) and all healthy controls (*n* = 6). Each cell type was divided into two groups, diseased and healthy controls, according to whether the donor had COVID-19. Subsequently, differential gene expression profiles between the diseased and healthy controls in specific cell-types were calculated by using the “FindMarkers” function in Seurat (Appendix A) [35]. GSEA analysis of repurposing drug-induced THP-1 differential expression genes (logFC > 1) and specific cell-type transcriptomes were used to assess the enrichment of sets of genes (repurposing drugs DE genes) in each cell type (scRNA seq gene list). For each repurposing drug, a specific cell with FDR < 0.05 and ES < 0 was identified as potential drug-sensitive cell types for COVID-19.

### 2.6. Network-Based Prediction of Drug Combinations

Drug combination therapies are more beneficial rather than individual drug since the synergistic drug pairs can target more genes and play role in multiple complicated pathways [36]. The Complementary Exposure model has previously been demonstrated as an effective approach to predict useful combinations [37]. The model is based on the following conditions: drug targets and disease genes overlap topologically $\left(Z_{\mathrm{DA}}<0,Z_{\mathrm{DB}}<0Z_{DA}<0,Z_{DB}<0\right)$, and two sets of drug targets are separated topologically $\left(S_{AB}>0\right)$. The Complementary Exposure model network proximity between a drug $\left(A\;\mathrm{or}\;B\right)$ and a disease  (D)  is defined by the z-score (Equation (3)):(3)$$Z_{DA}=\frac{d_{DA}-\mu_{d}}{\sigma_{d}}$$

The z-score is calculated by randomly sampling both degrees of nodes (drug targets and disease genes) with 1000 replications. The mean distance $\mu_{d}$ and standard deviation $\sigma_{d}$ of the reference distribution are used to convert $d_{DA}$ to a normalized distance (Equation (4)), where $d_{DA}$ relies on the average shortest path lengths $d\left(d,a\right)\;$ between disease genes $\left(d,d\in D\right)$ and drug targets $\left(a,a\in A\right)$.
(4)$$d_{DA}=\frac{1}{\left\|D\right\|}\underset{d\in D}{\sum }min_{a\in A}d\left(d,a\right)$$

The network-based separation $S_{AB}$ is quantified with two drug targets module *A* and *B* by calculating the mean shortest distances $d_{AA}$ and $d_{BB}$ (Equation (5)):(5)$$d_{AA}=\frac{1}{\left\|A\right\|}\underset{a\in A}{\sum }min_{a\text{'}\in A}d\left(a,a\text{'}\right)$$
where $a^{\prime }\;\left(a'\in A\right)$ is the closet node to $a\;\left(a\in A\right)$ within the interactome network. The mean shortest distance $d_{AB}$ between their proteins is defined by the “closest” measure, where $d\left(a,b\right)$ is the shortest path length between $a\;\left(a\in A\right)$ and $b\;\left(b\in B\right)$ in the interactome network (Equation (6)).
(6)$$d_{AB}=\frac{1}{\left\|A\|+\|B\right\|}\left(\sum_{a\in A}min_{b\in B}d\left(a,b\right)+\sum_{b\in B}min_{a\in A}d\left(a,b\right)\right)$$

A networked-based separation of a drug pair, *A* and *B*, can be calculated as follows (Equation (7)):(7)$$S_{AB}=\langle d_{AB}\rangle -\frac{\langle d_{AA}\rangle +\langle d_{BB}\rangle }{2}$$
where $d_{AB}=0$ if genes are included in both the drug *A* and *B* target modules [38].

## 3. Results

### 3.1. GO Enrichment Analysis of COVID-19 Related Genes

To obtain meaningful molecular mechanisms underlying COVID-19, GO enrichment analysis classified potential COVID-19 related genes into enriched terms (Appendix A). All 63 SARS-CoV-2 related genes were categorized functionally into 1035 Gene Ontology terms including biological processes, cellular components, and molecular functions. Among the 971 COVID-19 DEGs, 860 genes were enriched in 1399 Gene Ontology terms. The COVID-19 related genes we identified were significantly enriched in blood pressure regulation (*p*-value = 5.29 × 10^−23^), inflammatory response (*p*-value = 3.62 × 10^−09^), neutrophil activation (*p*-value = 6.16 × 10^−60^), and response to virus (*p*-value = 8.68 × 10^−32^) (Figure 2). The results are consistent with previous studies, indicating that the renin-angiotensin system (RAS) plays an important role in the biological mechanisms of COVID-19 [39,40].

### 3.2. Network-Based Proximity Scores between Drug–Target Modules and COVID-19 Related Genes

We obtained the drug–disease proximity scores to evaluate the drug effect on COVID-19 through a network-based calculation. Drugs with low proximity scores are more likely to be effective against SARS-CoV-2 infection since the proximity scores reflect the distance between drug target sets and COVID-19 related genes in the interactome networks. Using this approach, we explored the distance of 7811 drug–target modules and COVID-19 related genes. The distance distribution of the drug targets to COVID-19 related genes was in the range of −2.66 to 2.79, and both real drugs and simulated drugs were widely distributed near the point of 1.70 (Figure 3). A ranked list of the potential drugs was clearly distributed in the range of −2.66 to 0.99, suggesting that the targets of existing drugs were closer to the COVID-19 genes than the reference sets (simulated drugs). We selected a distance smaller than 0.99 as the threshold to screen the potential drug candidates for COVID-19, where the corresponding Z-score was approximately −2.33 after converting into the proximity value. Finally, 468 drugs with proximity less than −2.33 were included in further analyses (Appendix A).

### 3.3. GSEA Analysis of COVID-19 Related Genes in Drug-Induced Signatures

To further estimate the drug candidate’s efficacy on the disease and explore the underlying signaling pathways, we performed GSEA to examine their impact on the transcriptome of THP-1 cells. Since drugs were not fully matched between DrugBank and LINCS, some drugs were removed during the matching progress. In the total of 7811 drugs included in DrugBank and 377 from LINCS (THP-1 cell line), 112 drugs were matched by common name and 101 were matched by InChI Key (International Chemical Identifier Key). After removing overlaps, 131 drugs were included in both DrugBank and LINCS, 27 of which had low proximity scores (Z < −2.33) and were obtained for further GSEA.

We identified 18 drugs (FDR < 0.25 and ES > 0, Table 1) as potential therapeutic candidates since they significantly affected the expression of COVID-19 related genes in the mononuclear cells (Appendix A). These candidates included anti-viral agents (curcumin, dexamethasone, chloroquine), anti-diabetic agents (glibenclamide), analgesics (resveratrol), anti-convulsant (valproic acid), anti-cholesteremic agents (simvastatin), anti-carcinogenic agents (phenethyl isothiocyanate), anti-neoplastic agents (tretinoin), immunosuppressive agents (fostamatinib, atorvastatin, cyclosporine), anti-estrogen (tamoxifen), anti-hypertensive (nicardipine, nifedipine), anti-allergic agents (promethazine), and anti-cancer agents (orantinib, tipifarnib).

### 3.4. Repurposing Drugs Sensitivity in Specific Cell Type

Differential expression analyses in 7 cell types between COVID-19 patients (*n* = 7) and controls (*n* = 6) were performed based on the scRNA-seq data (Appendix A). According to the GSEA analysis, the DE genes of most repurposing drugs were enriched significantly in B cells (Table 2, Appendix A). CD14+ Monocytes Cells and Dendritic Cells also showed sensitivity to the repurposing drug treatment. None repurposing drug DE genes were significantly enriched for the Single-cell gene expression spectrum of NK Cells, CD8+ T Cells, CD4+ T Cells.

### 3.5. Identification of Synergistic Drug Combinations

Based on the Complementary Exposure model, we identified 153 drug combinations based on the 18 potential therapeutic candidates for COVID-19. Among these combinations, 123 drug pairs were excluded due to close drug–target modules $\left(S_{AB}<0\right)$, while 30 drug combination conformed to the Complementary Exposure Model and may therefore be effective in the treatment of COVID-19 (Table 3).

One notable potential drug combination was fostamatinib $\left(F\right)$ plus tretinoin $\left(T\right)$. Fostamatinib $\left(Z_{DF}=-3.68\right)$ and tretinoin $\left(Z_{DT}=-2.44\right)$ targets were both overlapped with the COVID-19 disease module, indicating that the drug combination might have a therapeutic effect on the disease. At the same time, the targets of fostamatinib and tretinoin were independent with network-based separation $\left(S_{FT}>0\right)$, and therefore fit the Complementary Exposure pattern (Figure 4a). We also used the Sankey diagram to represent the interactions among drug–target-disease (Figure 4b). Apart from the drug directly targeting COVID-19 related genes, un-targetable drug–disease effects were present due to the drug–target interaction with COVID-19 related genes in the PPI as reflected by the proximity scores. Additionally, take promethazine $\left(P\right)$ and nicardipine $\left(N\right)$ as a counterexample. Promethazine ($Z_{DP}=-2.58$) and nicardipine $\left(Z_{DN}=-2.81\right)$ targets fell into the Overlapping Exposure with the COVID-19 disease module. Although promethazine and nicardipine showed effective treatment on the disease, overlapping drug pair $\left(S_{PN}<0\right)$ was not a synergistic drug pair due to adverse effects such as overlapping drug toxicity (Figure 4c). Additionally, sharing targets of promethazine and nicardipine meant the drug pair had limits in treatment from different therapeutic pathways (Figure 4d).

## 4. Discussion

This study used a network-based drug repurposing combined with a transcriptomics strategy to identify potential drug candidates and drug pairs for COVID-19 treatment. The joint analysis of the proximity of drug–target relationship modules, SARS-CoV-2 genomics, transcriptomics, and synergistic drug effects could overcome the limitations of analyzing data from only network distance or transcriptome and improve drug candidate prediction. We proposed 18 drugs and 30 drug combinations including broad-spectrum antiviral agents, receptor antagonists, channel blockers, and renin-angiotensin system agents.

Some medications such as dexamethasone, chloroquine, curcumin [41], glyburide [42], tretinoin [43,44], cyclosporine [45,46], valproic acid [47], fostamatinib [48,49], atorvastatin [50,51,52], and phenethyl-isothiocyanate [53] have recently received major attention for the treatment of COVID-19 and have been validated by previous studies, supporting the reliability of our findings. Nicardipine, promethazine, orantinib, and tipifarnib have not previously been reported as potential treatments for COVID-19. Therefore, we will discuss these novel drug candidates in the following.

Nicardipine

With a similar structure to nifedipine (Z = −2.68), nicardipine (Z = −2.75) was initially developed to regulate high blood pressure as a dihydropyridine calcium channel blocker [54]. Nifedipine is indicated to potentially be effective in the treatment regimens of elderly patients with hypertension hospitalized with COVID-19 [55,56]. Therefore, nicardipine might play a similar role with nifedipine in the adjuvant treatment of COVID-19 patients.

Promethazine

Promethazine (Z = −5.65) antagonizes various receptors including dopaminergic, histamine, and cholinergic receptors, and is commonly used for indications such as allergic conditions, motion sickness, sedation, nausea, and vomiting [57]. The proximity score of promethazine was significantly low partly by targeting genes including CALM1, KCNS1, LPAR4, LPAR6, P2RY12, P2PY8, and P2RX5, which were DEGs between T cell subsets of COVID-19 samples and healthy controls. Characteristics of the bronchoalveolar immune genes have been explored as potential mechanisms underlying pathogenesis in COVID-19 [58]. These findings implied that promethazine might be effective for COVID-19 by regulating the immune cell microenvironment.

Orantinib and Tipifarnib

Orantinib (Z = −2.54) showed preliminary efficacy and safety in advanced hepatocellular carcinoma [59]. Tipifarnib (Z = −2.40) was studied in the treatment of acute myeloid leukemia (AML) and other types of cancer [60]. Although orantinib and tipifarnib are both not yet approved by the FDA, anticancer drugs identified by our study such as phenethyl isothiocyanate have been reported to be an effective treatment strategy to treat COVID-19 [53]. Drug repurposing against COVID-19 focused on anticancer agents was previously predicted to be effective and it was speculated that drugs interfering with specific cancer cell pathways may be effective in reducing viral replication [61]. Therefore, the anticancer drugs orantinib and tipifarnib might also be potential candidates for the treatment of COVID-19.

In contrast with our results, tamoxifen (Z = −4.75) was reported to increase the COVID-19 risk due to its anti-estrogen and P-glycoprotein inhibitory effects [62]. Data from previous experiments suggested that estrogen could regulate the expression of angiotensin-converting enzyme 2 (ACE2) [63], which was reported to be the critical natural cellular receptor for SARS-CoV-2 and was an important factor for infection. However, a recent study discussed the uncertain effects of RAS blockers on ACE2 levels and activity in humans and proposed an alternative hypothesis that ACE2 might more likely be beneficial than harmful in patients with lung injury [64]. The controversies of ACE2 system inhibition attempt to explain the relationship between the virus and the RAS [65], but existing research is too limited to support or refute these hypotheses. Our research suggested that tamoxifen may influence cytokine storm syndrome by regulating cytokine-mediated signaling pathways (ES = 0.67, P = 0.14), which is a severe clinical symptom of COVID-19 [66,67]. Several studies have indicated that tamoxifen could reduce cytokines to normal levels and it has been demonstrated to be beneficial for inflammation in rats [68,69]. Overall, we recommend that tamoxifen may protect against cytokine storms and alleviate ARDS in COVID-19 patients as well as reduce the incidence of critical illness and mortality.

There are some limitations to our strategy. First, the proximity calculation regards proteins interaction as nodes and links, which may not completely capture important information about the interaction types. Second, the LINCS and DrugBank databases are only partly matched, and therefore many important drug candidates may be ignored. Additionally, some of the potentially interesting drugs, such as alemtuzumab (Z = −3.27), were not able to be included in the final screening. Third, although THP-1 cells might be a useful tool in the research of monocyte and macrophage-related mechanisms [70], heterogeneity still exists in the gene expression profile of the mononuclear cells of COVID-19 patients and THP-1 cells. Additionally, considering that the impaired function of heart, brain, lung, and liver were complications of COVID-19 [71], more types of infection-related cell lines could be taken into account to fully investigate drugs and treatment outcome on COVID-19.

In conclusion, our effective drug repurposing strategy combined network-based pharmacology and transcriptomes methods to identify 18 potential COVID-19 drugs, and recommend 30 drug combinations. Although several candidate repurposing drugs were previously reported to have the anti-COVID-19 effect, four drugs such as nicardipine, promethazine, orantinib, and tipifarnib were recommended for the first time in COVID-19 treatment. Additionally, based on our repurposing drug sensitivity analysis, DE genes of most repurposing drugs were enriched significantly in B cells. Our analysis contributed to guide and accelerate research in COVID-19 drug development, and this method would be kindly applicable for drug repurposing research in future complex diseases. However, the identified drug candidates still require future experimental validation and large-scale clinical trials before their use in COVID-19 management.

## Figures and Tables

**Figure 1 pharmaceutics-13-00545-f001:**
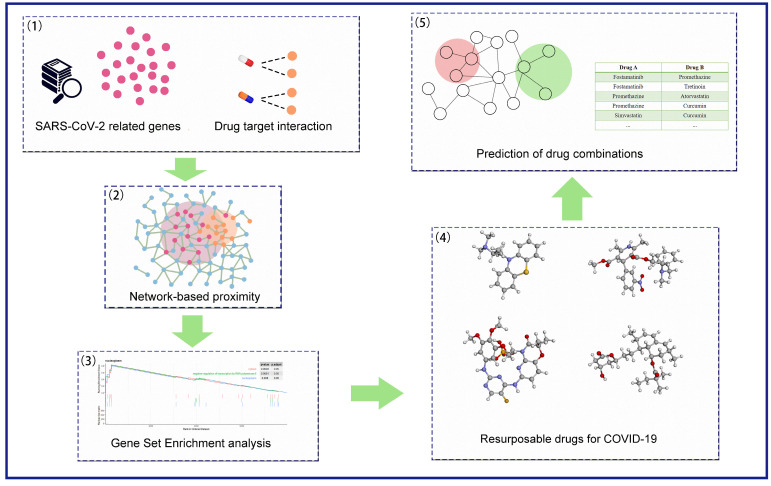
Schematic illustration of the computational framework. (1) Collection of the coronavirus disease 2019 (COVID-19) related genes from published SARS-CoV-2 human host data and differential expression genes (DEGs) from a single-cell study of the peripheral immune response in patients with severe COVID-19 (GSE150728). (2) Drug–target information retrieved from DrugBank and SuperTarget. (3) Quantify the therapeutic effect by computing the proximity between drug targets and COVID-19 related genes. (4) Gene set enrichment analysis (GSEA) to determine whether COVID-19 related genes show significance in drug-induced gene expression profiles. (5) Drug candidates were further prioritized for drug combinations using the “Complementary Exposure” model.

**Figure 2 pharmaceutics-13-00545-f002:**
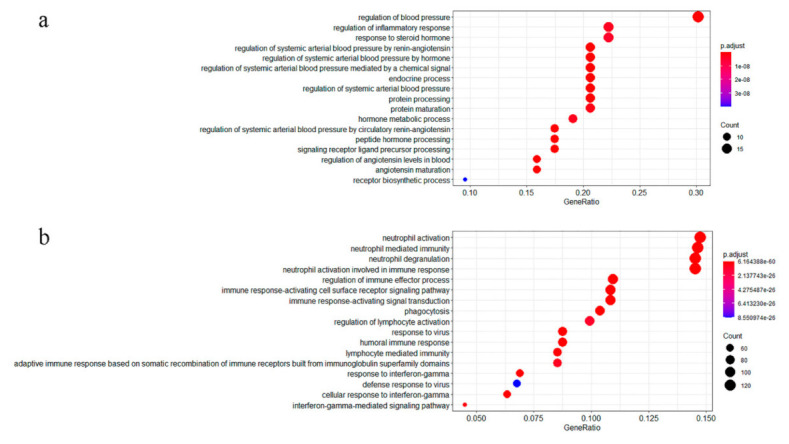
GO enrichment analysis of COVID-19 related genes. The dot plot is used to visualize enriched terms, (**a**) shows the COVID-19 related genes (*n* = 63) enrichment visualization and category interpretation. (**b**) pathway enrichment analysis visualization of single-cell DEGs (*n* = 860).

**Figure 3 pharmaceutics-13-00545-f003:**
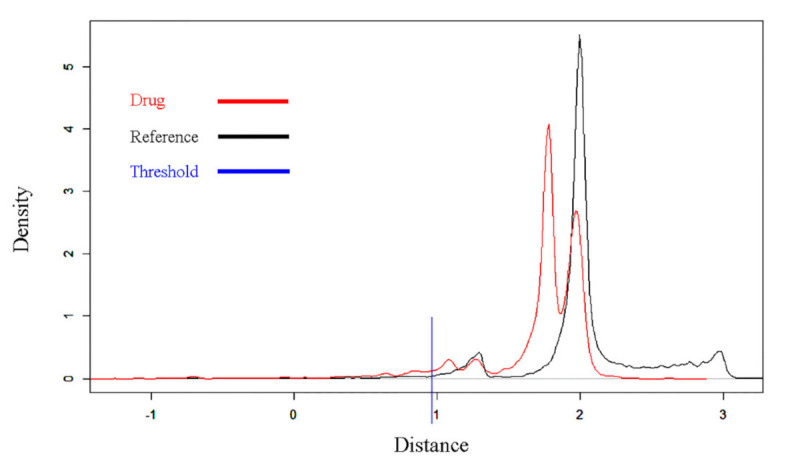
Distance distribution of all 7811 drugs and simulated reference. Peaks suggest that the distance corresponding to most members was around this value. The red line shows the distribution of the distance of the 7811 drugs to COVID-19 related genes. The black line illustrates the distance distribution of the simulated reference based on 30,000 replications. The blue line shows the threshold (distance < 0.99, Z-score < −2.33) to screen the drug candidates for COVID-19.

**Figure 4 pharmaceutics-13-00545-f004:**
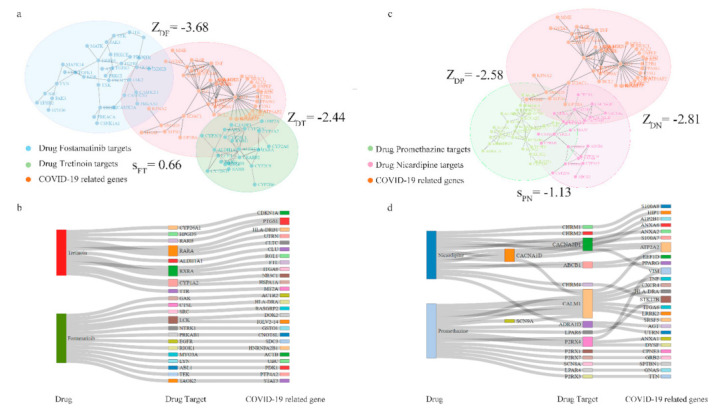
Network-based stratification of hypertensive drug combinations. (**a**) A network-based separation of a drug pair, fostamatinib (F), and tretinoin (T). For $Z_{DF}<0$ and $Z_{DT}<0$, the drug–target module of fostamatinib (F) and tretinoin (T) was overlapped with the disease module (D). For $S_{FT}>0$, the two sets of drug targets are separated topologically. Fostamatinib and tretinoin targets both separately hit the COVID-19 module, which was captured by the Complementary Exposure pattern. The disease module in orange (D) included disease-related genes (nodes) and their undirected and unweighted interactions (links), while the drug module (F or T) in blue (green) included drug–targets (nodes) and their undirected and unweighted interactions (links). (**b**) Sankey diagram visualizes drug pairs’ mechanism hypothesis: drugs are on the left, and COVID-19 related genes are right. Links show drugs that were mapped onto COVID-19 related genes through drug–target associations and human protein-protein interaction. (**c**) Nicardipine (N) and Promethazine (P) drug–target modules overlapped the network. For $S_{PN}<0$, the two sets of drug targets were Overlapping Exposure, which meant more adverse effects and less efficacy compared to the Complementary Exposure pattern. (**d**) Sankey diagram showed how drug–targets of Nicardipine and Promethazine overlapped and interacted with related genes.

**Table 1 pharmaceutics-13-00545-t001:** Eighteen repurposable candidates for COVID-19.

DrugBank ID	Z-Score	Drug Name	Structure	Pharmacodynamics	Reported Studies of COVID-19(PMID)
DB12010	−8.75	Fostamatinib	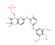	immunosuppressive agents	32637960
DB12695	−6.64	Phenethyl-isothiocyanate	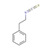	anti-carcinogenic agents	33131530
DB01069	−5.65	Promethazine	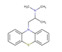	anti-allergic agents	NA ^1^
DB00641	−5.49	Simvastatin	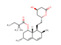	anti-cholesteremic agents	32626922
DB00675	−4.75	Tamoxifen	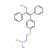	anti-estrogen	32663742
DB01076	−4.74	Atorvastatin	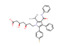	immunosuppressive agents	3266499032817953
DB11672	−3.65	Curcumin	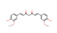	antiviral agents	3243099632442323
DB00755	−3.37	Tretinoin	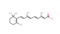	anti-neoplastic agents	32707573
DB01234	−3.21	Dexamethasone	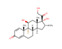	antiviral agents	3270655332620554
DB00608	−3.14	Chloroquine	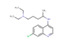	antiviral agents	3214536332147496
DB00313	−2.90	Valproic acid	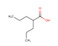	anti-convulsant	32498007
DB01016	−2.82	Glibenclamide	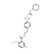	antiviral agents	32787684
DB00622	−2.75	Nicardipine	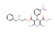	anti-hypertensive	NA
DB01115	−2.68	Nifedipine	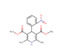	anti-hypertensive	3222669532411566
DB00091	−2.65	Cyclosporine	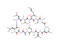	immunosuppressive agents	3237642232487139
DB02709	−5.63	Resveratrol	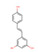	analgesics	3241215832764275
DB12072	−2.54	Orantinib	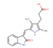	anti-cancer agents	NA
DB04960	−2.40	Tipifarnib	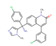	anti-cancer agents	NA

^1^ NA: Not previously been reported as potential treatments for COVID-19.

**Table 2 pharmaceutics-13-00545-t002:** GSEA analysis of drug-induced different expression (DE) genes in scRNA profiles.

Drug Name	B Cells	CD14+ Monocytes Cells	CD16+ Monocytes Cells	Dendritic Cells	NK Cells	CD4+ T Cells	CD8+ T Cells
Chloroquine	NA ^1^	NA	NA	NA	NA	NA	NA
Nicardipine	Significant ^2^	Significant	NA	NA	NA	NA	NA
Simvastatin	NA	NA	NA	NA	NA	NA	NA
Tamoxifen	Significant	Significant	NA	Significant	NA	NA	NA
Promethazine	NA	NA	NA	NA	NA	NA	NA
Nifedipine	Significant	NA	NA	NA	NA	NA	NA
Resveratrol	Significant	NA	NA	Significant	NA	NA	NA
Tipifarnib	Significant	Significant	NA	Significant	NA	NA	NA
Orantinib	NA	NA	NA	NA	NA	NA	NA
Tretinoin	Significant	Significant	Significant	Significant	NA	NA	NA
Atorvastatin	Significant	NA	NA	NA	NA	NA	NA
Dexamethasone	Significant	Significant	Significant	Significant	NA	NA	NA
Curcumin	NA	NA	NA	NA	NA	NA	NA
Fostamatinib	Significant	Significant	NA	Significant	NA	NA	NA
Valproic-acid	Significant	NA	NA	NA	NA	NA	NA
Glibenclamide	Significant	Significant	NA	NA	NA	NA	NA
Phenethyl Isothiocyanate	Significant	NA	NA	NA	NA	NA	NA
Cyclosporin	Significant	NA	NA	NA	NA	NA	NA

^1^ Significant: Drug-induced DE genes statistically significant enrichment in scRNA profile; ^2^ NA: Drug-induced DE genes statistically no significant enrichment in scRNA profile.

**Table 3 pharmaceutics-13-00545-t003:** All predicted possible combinations for COVID-19.

Drug A	Drug B	Drug A Common. Name	Drug B Common.Name	$S_{AB}$	$Z_{DA}$	$Z_{DB}$
DB01069	DB12072	Promethazine	Orantinib	0.76	−2.58	−2.53
DB12072	DB00313	Orantinib	Valproic acid	0.67	−2.53	−2.99
DB12072	DB00755	Orantinib	Tretinoin	0.66	−2.53	−2.44
DB00755	DB12010	Tretinoin	Fostamatinib	0.66	−2.44	−3.68
DB00622	DB12072	Nicardipine	Orantinib	0.60	−2.81	−2.53
DB01115	DB12072	Nifedipine	Orantinib	0.57	−2.71	−2.53
DB12072	DB01234	Orantinib	Dexamethasone	0.54	−2.53	−3.40
DB01069	DB04960	Promethazine	Tipifarnib	0.49	−2.58	−2.35
DB12695	DB00091	Phenethyl Isothiocyanate	Cyclosporine	0.43	−3.22	−2.67
DB04960	DB12695	Tipifarnib	Phenethyl Isothiocyanate	0.43	−2.35	−3.22
DB00675	DB12072	Tamoxifen	Orantinib	0.42	−3.40	−2.53
DB01069	DB12010	Promethazine	Fostamatinib	0.42	−2.58	−3.68
DB12072	DB01016	Orantinib	Glyburide	0.40	−2.53	−2.90
DB00641	DB12072	Simvastatin	Orantinib	0.39	−4.37	−2.53
DB12072	DB00091	Orantinib	Cyclosporine	0.37	−2.53	−2.67
DB02709	DB12072	Resveratrol	Orantinib	0.37	−3.91	−2.53
DB12072	DB01076	Orantinib	Atorvastatin	0.37	−2.53	−4.23
DB01069	DB01076	Promethazine	Atorvastatin	0.37	−2.58	−4.23
DB01069	DB12695	Promethazine	Phenethyl Isothiocyanate	0.34	−2.58	−3.22
DB00608	DB12072	Chloroquine	Orantinib	0.34	−3.31	−2.53
DB01069	DB02709	Promethazine	Resveratrol	0.33	−2.58	−3.91
DB12072	DB12695	Orantinib	Phenethyl Isothiocyanate	0.30	−2.53	−3.22
DB01069	DB11672	Promethazine	Curcumin	0.26	−2.58	−2.81
DB01016	DB12695	Glyburide	Phenethyl Isothiocyanate	0.18	−2.90	−3.22
DB12010	DB12695	Fostamatinib	Phenethyl Isothiocyanate	0.17	−3.68	−3.22
DB00622	DB12695	Nicardipine	Phenethyl Isothiocyanate	0.16	−2.81	−3.22
DB04960	DB00755	Tipifarnib	Tretinoin	0.14	−2.35	−2.44
DB01076	DB12695	Atorvastatin	Phenethyl Isothiocyanate	0.11	−4.23	−3.22
DB11672	DB12695	Curcumin	Phenethyl Isothiocyanate	0.06	−2.81	−3.22
DB00608	DB11672	Chloroquine	Curcumin	0.04	−3.31	−2.81

## Data Availability

Not applicate.

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
