# Peer review of "Drug Repurposing for COVID-19 Treatment by Integrating Network Pharmacology and Transcriptomics"

_pharmaceutics, 2021, doi:10.3390/pharmaceutics13040545_

Round 1

Reviewer 1 Report

The authors present a robust and straightforward strategy to evaluate drug and drug combo candidates for diseases, specifically for COVID in this manuscript. The authors leveraged the protein-protein interaction network as the background network to then evaluate covid-related genes (acquired from recent single cell covid studies) and their overlap/distance with drug-target subnetworks. This allowed for a quantitative strategy to prioritize drug combination candidates using data-driven (drug-target) and knowledge base driven (ppi) resources, which could be applied to any drug combo/subnetwork combo application. This manuscript is aptly suited for general interest of the broad readership and is deserving of publication after addressing the following comments:

Methods:

Section 2.1: After the first sentence, it is essential to add a bit more detail as to how the covid-19 DE gene list was assembled from this recent scRNAseq publication (GSE150728). Did you combine all cell-specific DE tables from their supplementary tables and use all unique genes to get the ~1000 genes? If so, please the specific supplementary tables/info that was used from this publication to assemble the list. It may also be worth mentioning in the methods how many genes came from each cell type and what cell types were included.

If it was generated de novo, it would be required to add your scRNAseq analysis method to this section.

Results

Section 3.3: After GSEA analysis, there only remained 18 drugs that passed the filtering criteria. It would be interesting to analyze the specific covid genes that were enriched in those 18 drugs gene lists, considering many of the setSizes (column D of Supp Table 4) are small. Specifically, please report how many total unique genes of the 800+ covid19 genes were enriched in all of the 18 drugs. For example, this could help answer if it is the same 20-50 genes that always show up that are involved in many different processes, and may allude to a common process. Also, it'd be worth reporting this same number for all drugs or the drugs that aren't significant at the given threshold. This would help provide a view of the general potentail of how many covid19 genes could be enriched. 

Related to this, given that the covid genes came from the single cell study that included myleoid cell types and considering the THP-1 cells influence on prioritization (as the authors rightly mention in discussion), it would be interesting to check to see if that aggregate list of "drug-enriched covid19 genes" from the 18 drugs are enriched for a specific cell type(s). This could be done using the author's method, by using the DE gene list per cell type (from the covid scRNAseq paper) as input lists (instead of LINC gene lists) for GSEA and use this newly assembled "drug-enriched covid19 genes" as the gene set. This would provide a general way to check the bias/influence, or lack there of, of myeloid cell types/THP-1 on the output of this data.

Section 3.4:  To help visualize the complementary exposure strategy, I recommend adding to figure 4 to show a module overlap network image (as in A) and a sankey diagram (as in B) of a low scoring and/or bad scoring combination. This would help the reader to see how "notable " the fostamatinib plus tretinoin potential drug combination may be.

Reviewer 2 Report

The authors has produced the article with excellent work

The following points should be considered before acceptance

  1. all figures need to be submitted in good quality.
  2. The introduction is lacking information about the GSEA analysis and its importance towards these drugs.
  3. The introduction could be modified on excluding the highlights of COVID-19 (everyone knows the pandemic disaster) instead please highlight the importance of drugs used in the study and should be discussed how important study can bring importance to health and biomedical researchers.
  4. The conclusion should be projected with more relevant additional information.
  5. Grammatical and spell check can be improved

Round 2

Reviewer 1 Report

I appreciate the authors for each of the attentive and detailed additions and responses, especially in regards to the additions of the counterexamples and data-driven cell-specific insights. This revised manuscript is deserving of publication in this journal.

Author Response

Thank you very much for your comments and suggestions.

Reviewer 2 Report

Can be accepted with this version

Author Response

(The authors gave the same response as above.)
